# Synthesizing Tabular Data with Latent Semantic Regularization

## Abstract

Modern generative models have shown remarkable capabilities in synthesizing tabular data, yet they often fall short in preserving the semantic integrity of generated samples, which can be interpreted as a form of hallucination. To address this gap, we propose a novel framework that formulates this problem as a constrained optimization problem and provides a solution for unsupervised learning of the implicit semantic constraints of the data and subsequently encouraging the generative model to respect the learned semantic boundaries through regularization. Our framework includes a *validator* component in form of a latent space model that is tasked with capturing the underlying semantic structures of the training data. This generic validator can be used to regularize the *synthesizer* model and steer it towards improving the semantic integrity of the synthesized data. We showcase our framework with a VAE-based validator and GAN-based synthesizer. We propose metrics designed specifically to measure the semantic integrity of the synthesized data and demonstrate that our approach not only maintains general quality of the generated data but also ensures a higher adherence to complex, domain-specific semantic relationships within the generated datasets.

## 1 Introduction

While training machine learning models on huge troves of data has many benefits (e.g. See (Kaplan et al., 2020)), it comes with its own challenges. Availability is one of these issues: physical constraints limit access to distributed big data. Privacy is another: data holders often are unable or unwilling to share their private datasets. One way to mitigate these problems is by utilizing synthetic data (Rubin, 1993)—where a proxy dataset is generated for downstream tasks such as data analysis and training *Machine Learning (ML)* models. Modern data synthesis is often done by a model, which can be viewed as a compressed representation of the data. This solves the data access problem since the compact model can generate an unlimited amount of data. Furthermore, privacy is preserved to some extent since the real, original data is not used for the downstream tasks.

We consider the task of synthesizing high-fidelity tabular data. Specifically, we focus on the challenge of ensuring the synthetic data is *semantically correct*. Semantically incorrect records have some combination of features which violates some semantic rule, either explicitly known or implicit. Egregious violations of inherent semantic rules within the dataset's domain can be interpreted as a form of *hallucination* in data synthesis by generation of synthetic records that represent implausible or impossible real-world scenarios. In mission critical domains such as healthcare, preserving semantic integrity is essential. For instance, it should be impossible to generate a patient record whose sex at birth is male and who is also pregnant.

We define the task of training generative models with enhanced semantic integrity as an unsupervised constrained optimization problem. We propose a framework for the unsupervised estimation of the implicit underlying semantic rules from the data and then using them to steer the generative model towards improving the semantic integrity of its output. We add a latent-space probabilistic *evaluator* component to a *synthesizer* generative model that would ensure the data is semantically valid by putting soft bounds on the latent space of the synthesizer. Specifically, we augment a *Generative Adversarial Network (GAN)* synthesizer with a *Variational AutoEncoder (VAE)* evaluator. The GAN is incentivized to explore its latent space to produce as

diverse a dataset as possible. Yet the VAE simultaneously applies soft semantic bounds to improve semantic integrity. We provide a unified optimization objective that balances this explore-exploit trade off. While we provide a specific choice of architecture for both the generative model and the evaluator component, we argue that this loosely-coupled framework could be applied and extended to a variety of latent space models for both, which will give researchers freedom in improving the semantic integrity aspects of their own data synthesis models.

Our contributions are:

- A framework for the task of generating high fidelity data with improved semantic integrity

- Improving semantic integrity through an unsupervised plug-in method

- Designing specific metrics to measure the quality and semantic integrity of the generated data

- Reporting superior performance compared to state of the art tabular data synthesizers

## 2 Semantic Integrity in Generative Models

Semantic integrity in the context of generative models refers to the property that ensures the generated data is semantically consistent with domain-specific knowledge and constraints. It ensures that the generated instances by a model are not only statistically plausible but also contextually coherent and in compliance with known domain-specific constraints and facts (e.g., biological, logical, or cultural norms).

The problem of enhancing semantic integrity in a generative model can be cast as an unsupervised constrained optimization problem. Let $f(x, z)$ be a latent variable generative model, where $x$ is a data point from the true data distribution $p_{\text{data}}(x)$, and $z$ is a latent variable sampled from some prior distribution $p(z)$. The model aims to learn the data distribution through $p_{\text{model}}(x)$ such that $p_{\text{model}}(x) \approx p_{\text{data}}(x)$. However, the assertion that $p_{\text{model}}(x) \approx p_{\text{data}}(x)$ is an idealization that often does not hold due to the intrinsic limitation of models in capturing the full complexity of the data distribution, especially in high-dimensional spaces where data can exhibit intricate manifolds, and the presence of model bias or underfitting can lead to discrepancies; this is compounded by the fact that $p_{\text{data}}(x)$ itself may only be partially observed or contain noise, making the true distribution elusive. Moreover, even if $p_{\text{model}}(x)$ statistically resembles $p_{\text{data}}(x)$ over the support of the data, it may fail to enforce semantic constraints that are not explicitly encoded in the data but are vital for maintaining the integrity of the generated outputs (such as physical laws in simulation or canonical forms in language). For this reason, higher-level checks are crucial to ensure that the generative model adheres to these semantic constraints, thereby achieving not only statistical fidelity but also semantic integrity.

We denote the unknown constraint as $C(x)$, which returns a measure of the semantic integrity of $x$. The unsupervised constrained optimization problem can be formulated as:

$$
\begin{aligned}
\underset{\theta}{\text{maximize}} \quad & \mathbb{E}_{x \sim p_{\text{data}}(x), z \sim p(z)}[\log p(x|z; \theta)] \\
\text{subject to} \quad & C(x') \text{ is satisfied for all } x' \text{ generated by } f(z).
\end{aligned}
\tag{1}
$$

The first line in Equation 1 is a lower bound on the expectation over $p(x; \theta)$, facilitating an approximate but tractable solution to an otherwise computationally prohibitive likelihood calculation. Solving this problem implies a balancing act between generation diversity and semantic integrity. The objective involves balancing exploration (diversity of $x'$) and exploitation (semantic integrity of $x'$)[1]. High exploration can lead to diverse samples but risks violating semantic constraints, while high exploitation can ensure semantic integrity but may result in low diversity.

Solving this unsupervised constrained optimization problem is inherently difficult due to:

- *Implicit Constraints:* Semantic integrity constraints $C(x)$ are typically not explicitly defined or easily quantifiable, making it difficult to integrate them into the optimization framework.

---

[1]We will use a GAN for $f(z)$, and a divergence function for $C$, both explain later in this section.

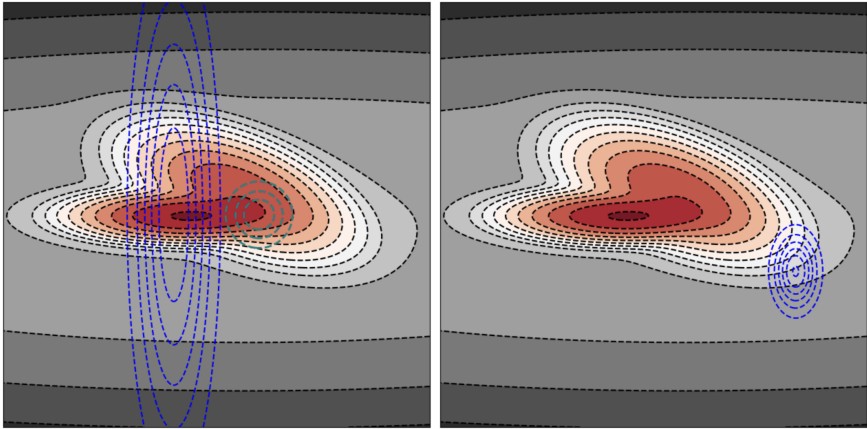

Figure 1: **Left**: Explore/exploit dilemma of synthesizing data under semantic constraints. *Green*: A model inclined towards generating samples with high semantic integrity would be concentrated around the mean of the distribution and will not capture the tail behavior of the latent space, leading to suppression of minority classes and poor exploration. *Blue*: A model optimized for diverse generation of samples would overestimate (some of the) tails of the target distribution, leading to the generation of semantically incorrect samples. **Right**: empirical posterior of a minibatch generated by $f$ and mapped to the validator latent space.

- *Complex Domains:* Real-world domains can have complex rules; ensuring semantic integrity could mean navigating a convoluted manifold with intricate boundaries separating valid from invalid regions.

- *Diversity vs. Constraint Satisfaction:* Balancing diversity with semantic integrity is a non-trivial trade-off.

- *Intractability in x space:* Enforcing semantic integrity constraints directly in the $x$ space is difficult due to it being sparse or pathological and $C(x)$ involving non-linear and non-convex conditions.

- *Heavy-Tailed, Latent Representation:* Manifolds in the latent space where data representations are situated could exhibit heavy tails. This is exacerbated by the curse of dimensionality.

## 3 Enforcing Soft Bounds on Latent Space to Improve Semantic Integrity

**Method Overview**  To enhance the semantic integrity of the model output in an unsupervised manner, we propose enforcing soft bounds on the latent space of the generative model. By imposing bounds on the latent space, we directly regulate the range of possible feature combinations that the model might generate in a non-deterministic manner. This allows us to prevent the model from producing samples that violate semantic constraints or fall outside of the expected data domain while encouraging the model to explore the support of the joint distribution. Realizing these constrains is particularly important for tabular data, where different modalities may have complex dependencies and tail behaviors. Since we make no assumptions about the structure and distribution of the latent space and its tail behavior, enforcing soft bounds helps mitigate the risk of mis-estimating the tail of the target distributions which would lead to either losing diversity of samples or generating semantically incorrect ones.

While other unsupervised methods are architecture specific (e.g. see Notin et al. (2021)), we propose an architecture-agnostic hybrid approach, with two generative models as ***synthesizer*** and ***validator***. The task of the synthesizer is to estimate a joint distribution over the input data features, while the validator learns and enforces soft boundaries on the latent space of the synthesizer. The synthesizer and the validator could potentially be any latent space generative model.

**Formal Definition**  Formally, consider the synthesizer model $f(x, z; \theta)$ defined in Section 2 that approximates the true data distribution $p(x)$ and generates samples $x'$. The synthesizer is a function of observed

data $x$ and latent variables $z \in Z$, which is a vector-valued variable introduced in the synthesizer model $f$ to capture hidden stochastic processes that govern the generation of data samples $x'$ from an approximated data distribution $p(x)$. The latent space $Z$ encapsulates unobserved characteristics of the observable space $X$ and is typically inferred through a probabilistic framework by optimizing the parameters $\theta$ of the synthesizer model to maximize the likelihood of the data given the latent variables, $p_\theta(x|z)$, while simultaneously adhering to semantic constraints encapsulated by a constraint function $C$ (Equation 1). Semantic integrity is violated when the model generates samples that are semantically incorrect, despite being within the support of $p(x)$. To enforce semantic integrity, we introduce a secondary independent latent variable model $q(x, z'; \phi)$ as the validator, parameterized by $\phi$, trained to infer the posterior distribution $q_\phi(z'|x)$ over the latent variables given the training data. The validator $q$ captures the underlying semantic structure of the data implicitly via its latent space and is designed specifically to capture the tail behavior of the latent variable $z$ accurately. This allows us to push the synthesizer towards exploring the tail of the target distribution without violating its constraints. Specifically, our unsupervised constrained optimization problem can be expressed as:

$$\underset{\theta}{\text{minimize}} \; \mathbb{E}_{x \sim p(x), z \sim p(z)} \left[ -\log p_\theta(x|z) + \lambda_c C(x') \right] \tag{2}$$

where $C : X \to [0, \infty)$ is a constraint function that quantifies the semantic integrity of generated samples $x'$, where $X$ denotes the space of possible data samples. A value of 0 for $C$ indicates that the generated sample perfectly satisfies the semantic constraints. The optimization of $f$ is then guided by the learned posterior of $q$, through the minimization of the cross-entropy between $q_\phi(z'|x)$ and the prior $p_\theta(z')$. This acts as a Bayesian regularizer for the latent space of the synthesizer, encouraging the generative model to produce samples that are consistent with the semantic structure learned by the validator. Intuitively, the posterior $q_\phi(z'|x)$ serves as a probabilistic buffer, encapsulating the semantic constraints, while the cross-entropy term ensures that these constraints shape the latent space of $f$. During optimization, the parameters $\theta$ of the synthesizer model $f$ influence the generated samples $x'$ and, consequently, the value of $C(x'; \theta)$, thereby linking the constraint function to the model parameters.

Our proposed method is an unsupervised technique, which makes it particularly suitable for scenarios where explicit semantic rules are difficult or impossible to extract. The example we provided earlier of male patients with pregnancy history as an example of semantically incorrect samples violates a simple semantic bond, but one can imagine that complex interactions between different features of the data are either difficult to implement and use for labelling or in cases not explicitly discovered yet.

**Learned Priors vs. Ensembles**    The reason we use independent validator model(s) to enforce semantic soft bounds - as opposed to plug the learned latent model directly into the synthesizer - follows the ideas around effectiveness of anomaly and out of distribution detection using ensemble methods (e.g. Choi et al. (2018)). When the validator is separate, it can act as an objective evaluator, not influenced by the generation process of the synthesizer. This separation is critical for preventing overfitting, where a synthesizer could potentially learn to exploit any shared structure with the validator to minimize apparent error, rather than genuinely improving semantic integrity.

In the rest of this section we define a VAE validator and a GAN synthesizer and explain the semantic integrity feedback loop from validator to synthesizer. Furthermore, the use of two distinct models allows for specialization in their respective tasks. The synthesizer focuses on data generation, exploring the latent space for diversity, while the validator is specialized in understanding and enforcing semantic integrity. A separate validation process also allows fine-grained semantic integrity control by employing multiple class-conditional validator modules. A class-conditional validator enables feature-specific validation, which is advantageous in complex data domains where different features or feature sets may have varying levels of semantic complexity or importance. By adjusting the strictness or leniency of the semantic integrity bounds for each validator individually, the model can better accommodate domain-specific knowledge and constraints. This flexibility is particularly crucial in domains with disparate and nuanced semantic rules, allowing for tailored enforcement that aligns with the specific integrity requirements of each feature set.

### 3.1 Validator Component

For the validator component, we use a Conditional Variational Autoencoder (CVAE) (Sohn et al., 2015), which conditions the latent variable $\mathbf{z}$ on variable $\mathbf{y}$, with the objective function

$$\ln p(\mathbf{x} \mid \mathbf{y}) = \mathbb{E}_{\mathbf{z} \sim q_\phi(\mathbf{z}|\mathbf{x},\mathbf{y})}[\ln p(\mathbf{x} \mid \mathbf{z}, \mathbf{y})] - KL\left[q_\phi(\mathbf{z} \mid \mathbf{x}, \mathbf{y})\|p(\mathbf{z})\right].$$

VAEs are a good choice for the validator since their latent representation is structured and they allow approximate density estimation.

**Decomposition of ELBO Regularization Term**   To train a VAE, we maximize the Evidence Lower Bound (ELBO):

$$\ln p(\mathbf{x}) \geq \mathbb{E}_{\mathbf{z} \sim q_\phi(\mathbf{z}|\mathbf{x})}[\ln p_\theta(\mathbf{x} \mid \mathbf{z})] - KL\left[q_\phi(\mathbf{z} \mid \mathbf{x})\|p(\mathbf{z})\right]$$

The KL divergence between the variational posterior $q_\phi(\mathbf{z} \mid \mathbf{x})$ and the prior $p(\mathbf{z})$ can be expressed in terms of entropy and cross-entropy:

$$KL\left[q_\phi(\mathbf{z} \mid \mathbf{x})\|p(\mathbf{z})\right] = \int q_\phi(\mathbf{z} \mid \mathbf{x}) \ln \frac{q_\phi(\mathbf{z} \mid \mathbf{x})}{p(\mathbf{z})} d\mathbf{z} = -\mathbb{H}[q_\phi(\mathbf{z} \mid \mathbf{x})] + \mathbb{E}_{q_\phi(\mathbf{z}|\mathbf{x})}[\ln p(\mathbf{z})] \qquad (3)$$

Maximizing the ELBO maximizes the entropy of the variational distribution (encouraging it to be more spread out or uncertain) and minimizes the cross-entropy between the variational distribution and the prior (encouraging the variational distribution to be similar to the prior).

**Enhancing Exploitation by Utilizing Cross Entropy to Inform the Synthesizer**   To increase semantic integrity of the synthesizer, we leverage the learned prior of the validator by incorporating a feedback loop from the validator to the synthesizer to inform the generation process. Looking at the decomposition of KL term in 3, we posit that the entropy term does not take into account the prior $p(\mathbf{z})$ and therefore does not measure how well the distribution of generated samples fits our prior beliefs about the data. As such, minimizing entropy alone would lead to the encoder's output distribution $q_\phi(\mathbf{z} \mid \mathbf{x})$ being deterministic - which is not desirable if we want to increase the variety of the in-distribution generated samples- but would not guarantee that this determinism aligns with our prior knowledge encapsulated in $p(\mathbf{z})$. On the other hand, the cross-entropy term, $\mathbb{E}_{q_\phi(\mathbf{z}|\mathbf{x})}[\ln p(\mathbf{z})]$, reflects the expected likelihood of the latent variables $\mathbf{z}$ under the prior distribution $p(\mathbf{z})$. It quantifies the divergence of the encoder's distribution $q_\phi(\mathbf{z} \mid \mathbf{x})$ from our prior distribution $p(\mathbf{z})$. Therefore, minimizing the cross-entropy is akin to maximizing the posterior likelihood of the synthesized data informed by the VAE prior.

**Enhancing Exploration by Flexible Tail-Adaptive Prior**   A standard VAE typically employs a simple prior, such as a Gaussian distribution $p(\mathbf{z}) = \mathcal{N}(\mathbf{z}; \mathbf{0}, \mathbf{I})$, for the latent variable $\mathbf{z}$. However, this assumption may be too restrictive for modeling complex data distributions. To enhance the expressiveness of the VAE, a more flexible prior can be used. To this end, we use a tail-adaptive normalizing flow to capture the prior.

A normalizing flow prior defines a learnable complex distribution $p_\theta(\mathbf{z})$ by transforming a simple base distribution $p_0(\mathbf{z}0)$, which is typically a standard Gaussian, using a sequence of invertible functions parameterized by $\theta$. The latent variable $\mathbf{z}$ is thus obtained by applying the transformation $f\theta$ to the initial sample $\mathbf{z}_0$ drawn from $p_0(\mathbf{z}_0)$.

The ELBO for a VAE with a normalizing flow prior can be written as:

$$\ln p(\mathbf{x}) \geq \mathbb{E}_{\mathbf{z}_0 \sim p_0(\mathbf{z}_0)}\left[\ln p(\mathbf{x}|f\theta(\mathbf{z}0))\right] - KL\left[q\phi(\mathbf{z}|\mathbf{x})|p\theta(\mathbf{z})\right], \qquad (4)$$

where the Kullback-Leibler divergence term $KL\left[q\phi(\mathbf{z}|\mathbf{x})|p_\theta(\mathbf{z})\right]$ measures the discrepancy between the variational distribution and the learnable complex prior distribution.

The use of a normalizing flow as a prior allows for a more expressive posterior distribution by enabling it to capture complex behaviors. The change of variables formula necessitates the computation of the Jacobian determinant of the transformation $f_\theta$, which is given by $\ln \left| \det \frac{\partial f_\theta}{\partial \mathbf{z}_0} \right|$. This term corrects for the change in volume induced by the invertible transformation and ensures that the probability density remains normalized. For Conditional VAEs (CVAEs) (Kingma et al., 2014), which condition the latent variable $\mathbf{z}$ on another variable $\mathbf{y}$, the modified objective function with a normalizing flow prior is

$$\ln p(\mathbf{x} \mid \mathbf{y}) = \mathbb{E}_{\mathbf{z}_0 \sim p_0(\mathbf{z}_0)} \left[ \ln p(\mathbf{x} \mid f_\theta(\mathbf{z}_0), \mathbf{y}) \right] - KL \left[ q_\phi(f_\theta(\mathbf{z}_0) \mid \mathbf{x}, \mathbf{y}) \| p_0(\mathbf{z}_0) \right] + \ln \left| \det \frac{\partial f_\theta}{\partial \mathbf{z}_0} \right|, \qquad (5)$$

where the conditioning on $\mathbf{y}$ is applied throughout the model's generative process and the variational posterior.

Although using a flexible prior in validator helps us adhere to the implicit semantic structure of the data, the fact that most real-world data have mixed and heavy-tailed behavior could still limit the functionality of the model by either underestimating the tail which would hurt exploration or overestimating the tail which would violate semantic integrity. To fix this issue, we use a tail adaptive normalizing flow (anonymous, 2024) as the trainable prior of the validator. This flexible tail-adaptive prior allows the model to learn a complex, data-specific $p(\mathbf{z})$ and its tail behavior for the latent space. We interpret the learnable prior $p(\mathbf{z})$ as representing our beliefs about the distribution of semantic features in the latent space before observing any specific data. The encoder distribution $q_\phi(\mathbf{z} \mid \mathbf{x})$ is the posterior distribution which is our updated belief about where in the latent space a particular sample $\mathbf{x}$ is likely to be found, given the data we've observed.

### 3.2 Synthesizer Component

The data synthesizer in our method is a conditional WGAN-GP(Gulrajani et al., 2017). To encourage the synthesizer to explore the area between class boundaries further, we adopt an with an explicit auxiliary classifier following the AC-GAN architecture (AC)(Gong et al., 2019) and add the AC's negative entropy term to the loss function of the GAN which increases the entropy of the model. We make this change confidently since the validator component will reign in the synthesizer and keep it within the acceptable semantic boundaries.

The AC-GAN objective function can be defined as

$$\mathcal{L}_g = \mathbb{E}_{\mathbf{x} \sim \mathbb{P}_{Data}} [D(\mathbf{x} \mid \mathbf{y})] - \mathbb{E}_{\tilde{\mathbf{x}} \sim \mathbb{P}_{Gen}} [D(\tilde{\mathbf{x}} \mid \mathbf{y})] - \lambda_c \left( \mathbb{E}_{\mathbf{x} \sim \mathbb{P}_{Data}} [A(\mathbf{x} \mid \mathbf{y})] + \mathbb{E}_{\tilde{\mathbf{x}} \sim \mathbb{P}_{Gen}} [C(\tilde{\mathbf{x}} \mid \mathbf{y})] \right)$$

with $A$ being the implicit classifier and $\lambda_c$ the scaling factor.

The loss term of the auxiliary classifier is defined as

$$\mathcal{L}_{ac} = -\mathbb{E}_{\hat{\mathbf{x}} \sim \mathbb{P}_{\tilde{\mathbf{x}}}} \left[ \left( \|\nabla_{\hat{\mathbf{x}}} D(\hat{\mathbf{x}} \mid \mathbf{y})\|_2 - 1 \right)^2 \right].$$

To make the GAN training more robust, we also employ information loss Park et al. (2018) in our objective function. Information loss term matches the first and second-order statistics of $\mathbf{x}$ and $\hat{\mathbf{x}}$ from the last layer before the activation of the discriminator network, $\mathbf{v}$ as

$$\mathcal{L}_i = \left\| \boldsymbol{\mu}_{x \sim \mathbb{P}_{Data}} [\mathbf{v_x}] - \boldsymbol{\mu}_{\tilde{x} \sim \mathbb{P}_{Gen}} [\mathbf{v_{\tilde{x}}}] \right\|_2 + \left\| \boldsymbol{\sigma}_{x \sim \mathbb{P}_{Data}} [\mathbf{v_x}] - \boldsymbol{\sigma}_{\tilde{x} \sim \mathbb{P}_{Gen}} [\mathbf{v_{\tilde{x}}}] \right\|_2$$

with $\boldsymbol{\mu}[.]$ and $\boldsymbol{\sigma}[.]$ representing the mean and standard deviation of the feature vector $\mathbf{v}$.

**Regularized Synthesizer for Enhanced Semantic Integrity**   To increase semantic integrity of the data generated by the synthesizer as explained earlier in this section, we add the negative VAE cross entropy of the samples generated by GAN generator to its loss function by feeding the generated samples through the VAE encoder and feeding the resulting negative cross entropy to the GAN generator objective function. A higher cross-entropy means that the encoder distribution deviates significantly from the prior, implying

that the generated sample $\mathbf{x}$ is unlikely under our prior beliefs about the latent space. The validation error (Analogous to the constraint $C$ of Equation 2) is defined as:

$$C = \mathbb{E}_{q_\phi(\mathbf{z}|\mathbf{x})}[\ln p(\mathbf{z})]$$

The final objective function of the model is defined as

$$\min_{G,C} \max_{D} V(D,G) = \mathcal{L}_g + \mathcal{L}_i + \lambda_{ac}\mathcal{L}_{ac} + \lambda_c C$$

with $\lambda_{ac}$ and $\lambda_c$ being scaling factors for their respective loss terms.

## 4 Previous Works

This work proposes a specific method to put semantic constraints on a generative model in order to improve the semantic quality of its output. In order to achieve this goal, we propose a framework with two agnostic components: a generative model as a synthesizer, which estimates the joint and marginals of the data; and a validator component which learns semantic constraints in an unsupervised fashion and applies them as soft bound to the synthesizer. This requires combining two different generative models. We explicitly choose a VAE as our validator and a GAN as our generator to implement this framework and evaluate it.

In the literature the integration of various generative models to address specific limitations and enhance the functionality of these models has been extensively explored. For example, VAEs have been shown to suffer from inability to learn marginal distributions (Rosca et al., 2018). Thus they have been paired with other generative models to overcome these issues, a popular choice among them being GANs. Dumoulin et al. (2016) and Donahue et al. (2016) combine GANs and VAEs to improve performance on discriminative tasks. Makhzani et al. (2015) and Tolstikhin et al. (2017) transform the KL term into a synthetic likelihood representation that can be imposed by a discriminator and Bhattacharyya et al. (2019a) propose a "Best-of-Many-Samples" reconstruction loss term to further improve on fixing the problem of mode collapse and sample quality. Bao et al. (2017) propose combining a GAN and a CVAE to have more fine-grained control over the sample generation process. However, none of these methods, to the best of our knowledge, are aimed at explicitly defining and improving the semantic integrity of the generated data. Furthermore, our formulation of learning and imposition of semantic boundaries by the validator on the synthesizer leads to a lot of flexibility in choice of the model for each component and the impact of the validator on the synthesizer, differentiating it from the existing methods in the literature.

Combining normalizing flows and latent space models such as VAEs also has precedent in the literature for various goals. Ziegler & Rush (2019) combine normalizing flows and VAEs to overcome the problem of modelling discrete sequences and Bhattacharyya et al. (2019b) combine conditional flows and VAEs for structured sequence prediction. However, our purpose for modelling the validator prior using a tail-adaptive flow is specifically to capture the joint tail behavior of the prior, which has a significant impact on the semantic integrity of the synthesizer as we use the validator to guide it — encouraging the generation of more extreme cases in presence of heavy tails and vice versa.

Previous works comparable with our method explore improving semantic integrity of the generative models. While integration of explicit constraints has been previously studied (Stoian et al., 2024), we consider the task of implicit latent constraints, where the constraints are unknown and need to be both learned and applied to the generative model.

Park et al. (2018) propose a tabular data generation method based on Deep Convolutional GAN (DCGAN). They add a classifier to the model, with its loss added to the generator objective function - alongside an information loss term - to increase semantic integrity of the generated records. However, they don't provide an examination of the semantic integrity gain of their method. In this work, we provide an examination of the semantic integrity of this method along a select group of widely cited and used methods for comparison with our proposed method. Zhao et al. (2021) also use auxiliary classifier to train a GAN for tabular data generation. Engelmann & Lessmann (2021) use an auxiliary classifier to train a conditional WGAN. However, none of these methods explicitly mention using the auxiliary classifier to elevate the semantic integrity of the generated samples.

## 5 Experiments

To evaluate the performance of our proposed model, we compare its performance to three prominent tabular data generation methods, MedGAN (Choi et al., 2017), CTGAN (Xu et al., 2019) and TableGAN (Park et al., 2018). We use standard implementation of these three models, and we train all models on the same stratified samples of corresponding datasets for training and testing. We report the average values for five experiments and their standard error. For each experiment, we also report the results of ablation experiment of training only our synthesizer without the validator to present the impact of the validator both on general performance and semantic integrity.

We start with a motivating example of using a 2d dataset to showcase the impact of guiding the synthesizer to improve semantic integrity. Next we move on to real datasets and present general and statistical metrics to showcase the general performance of our method and how it will not be sacrificed in favor of improved integrity. Having established the general utility of our method in comparison with the other tabular synthesizers, we move on to the semantic quality of the generated samples by presenting exploration and exploitation metric results.

### 5.1 Metrics

We use average Kolmogorov-Smirnov and Chi-Square test scores as indicators of each model's general performance. We also use the following additional metrics to measure general and semantic performance of each model.

**Efficacy of Downstream Tasks**  The efficacy of synthetic datasets, $\mathcal{D}_g$, for downstream ML tasks is gauged via machine learning efficacy metrics. These metrics assess the performance parity of a classifier $f$ when trained on $\mathcal{D}_g$ relative to $\mathcal{D}_o$. The goal is to achieve statistical congruence between $\mathcal{D}_o$ and $\mathcal{D}_g$ without direct duplication of samples. The efficacy is measured by F-scores.

**Exploration Factor**  The exploration factor is defined as $S_d = \sum_{r=1}^{R} d_H(r, \hat{r})$, where $d_H$ is the adjusted Hamming distance between a row $r$ in the sampled data and its closest counterpart $\hat{r}$ in the original dataset. The adjustment of $d_H$ is predicated on the inclusion of only highly correlated categorical attributes, identified through a correlation matrix $\rho$, and selecting attribute combinations with $\rho_{ij} > \bar{\rho} \cdot \alpha$, where $\bar{\rho}$ is the mean of all correlations and $\alpha$ is a scaling factor set to 100 in this study.

**Semantic Integrity**  Semantic integrity in synthetic data lacks a universally accepted metric for assessment. Measuring semantic integrity is inherently complex, particularly in unsupervised contexts and across diverse dataset types and domains, often necessitating extensive domain expertise. To address this, we introduced synthetic features into datasets enabling controlled measurement of semantic integrity. We designed synthetic features to establish underrepresented and overrepresented subgroups within each feature. This was aimed at examining whether the generative model avoids producing semantically incorrect features by suppressing underrepresented classes and amplifying overrepresented ones.

### 5.2 General Experiment Setup

For our framework we use a CVAE as the validator, which is conditioned on all categorical variables to model the most challenging scenario with a focus on maintaining high semantic integrity throughout the dataset - assuming that in most real world use-cases the domain expert would be interested in preserving the semantic integrity of a subgroup of the dataset variables. The architecture of the CVAE includes an encoder and decoder, each comprising three fully connected layers with 128 neurons. The latent space dimensionality of the CVAE is set to 200.

The prior distribution for the CVAE is a Tail Adaptive Normalizing Flow anonymous (2024) with a depth of 4. We use a Real NVP (Dinh et al., 2016) architecture. The base density is modelled as a trainable mixture of 20 generalized Gaussian components and employs robust gradient estimation for training stability. Scale and shift operations of the RNVP are implemented using MLPs with a single hidden layer of 128 neurons.

Our synthesizer GAN has two MLPs as its generator and critic, each consisting of 2 layers of 256 neurons. For comparative baselines, we utilize the standard implementations of CTGAN, MEDGAN, and TABLEGAN as specified in their original publications. We train the validators and the synthesizers for 300 epochs.

Optimization is conducted using the ADAM optimizer with a learning rate of 2e-4. Our datasets are partitioned into an 80/20 percent train-test split. We repeat our experiments five times to ensure reliability, reporting both the average results and standard errors.

### 5.3   Motivating Experiments

**Data**   The toy dataset is sampled from a mixture of four two-dimensional Gaussians with one mixture labeled as 1 and the remaining three labeled as 0. The goal is to measure the general ability of the model to estimate the target distribution as well as generate correct labels. Each model was trained for 300 epochs. We provide accuracy of the generated labels as our quantitative metric.

**Experiment Setup**   For this experiment, we create a dataset by sampling from a mixture of three isotropic Gaussians and apply a clustering algorithm to label the clusters. These labels are added to the dataset, which is then used to train the models. To evaluate the semantic integrity of the data generated by the models, we use the original trained clustering model to assign labels to these synthetic data points. We label the samples are correct or incorrect based on whether the generated and predicted labels match. We report the percentage of samples with incorrect labels as an indicator of the generated datasets semantic integrity.

**Results**   Table 1 reports the percentage of incorrect labels generated by each model. We observe that our method performs best in terms of the number of samples with the wrong cluster label assignment. We also see a 1.5% degradation in performance of our synthesizer in the ablation experiment, motivating the impact of the validator.

**Efficacy of the Validator**   To showcase the efficacy of our proposed method in capturing the latent semantic structures, we train our model on this dataset with two validator VAEs, one with a Gaussian prior and one with our tail-adaptive normalizing flow prior. We investigate the efficacy of two validators for maintaining semantic integrity of generated data. The results are visualized in Figure 1, with samples with incorrect labels mapped to the latent space. The Gaussian prior VAE scatters semantically incorrect samples (marked as red dots) throughout the latent space, indicating a lack of structural awareness to differentiate between semantically correct and incorrect instances. In contrast, the VAE with the normalizing flow prior shows a more structured latent space, with incorrect samples predominantly clustering in the lower density regions, away from the core distribution of semantically correct data (the contour lines represent the density of the prior). This evidence points towards the conclusion that a flexible, tail-adaptive prior can potentially capture the semantic structures within the latent space.

### 5.4   Real Dataset Experiments

**Data**   In our real-world data experiments, we evaluate semantic integrity using several benchmark datasets: *Adult Census Income*, *Covertype* and *Census* from UCI Machine Learning Repository Dua et al. (2017), *Intrusion Detection* Tavallaee et al. (2009) and " Child Spiegelhalter (1992). These datasets span a range of applications, from income prediction to network intrusion detection, providing a comprehensive testbed for our experiments.

**General Performance Results**   Comparative analysis of the KS (Table 2) and Log Chi Square (Table 3) test results demonstrates the general efficacy of our proposed method. The KS test results indicate that our method achieves the lowest discrepancy scores for Adult, Intrusion and Census, suggesting superior model fidelity in capturing the underlying distributions of the real data. Notably, for the CovType dataset, TABLEGAN displays the best performance (0.11), however, our method remains competitive (0.12). The Log Chi Square test results corroborate these findings, with our method reporting the lowest error on the Census and Child datasets , although it shows higher error for CovType. When considering the ablation variant,

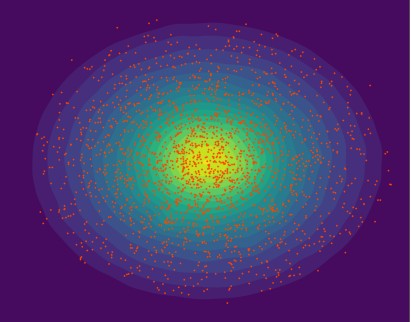 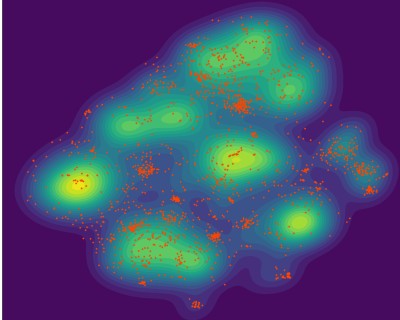

Figure 2: Comparative latent space visualization of semantically incorrect samples. On the left, a VAE validator with a Gaussian prior disperses the incorrectly labelled samples (red dots) throughout the latent space. On the right, the same VAE with a tail adaptive normalizing flow prior demonstrates semantically incorrect samples clustered within the low-density regions, better distinguishing semantically incorrect samples from regular data points. Contour lines represent the density of the prior distribution over the latent space.

Table 1: The percentage of the incorrectly labelled 2d samples.

| Model | Incorrect Labels ↓ |
|---|---|
| MEDGAN | 51.4% ± 11.9% |
| TABLEGAN | 44.2% ± 3.5% |
| CTGAN | 10.9% ± 0.5% |
| **Ours (S+V)** | **6.4% ± 0.1%** |
| Ours (S) | 7.9% ± 0.2% |

Table 2: KS test results for real datasets. We report the avarege KS test statistics for all numerical features of each dataset.

| Model | Adult ↓ | CovType ↓ | Intrusion↓ | Census ↓ | Child ↓ |
|---|---|---|---|---|---|
| MEDGAN | 0.92 ± 0.003 | 0.42 ± 0. | 0.18 ± 0.002 | 0.82 ± 0.005 | N/A |
| TABLEGAN | 0.40 ± 0.02 | **0.11 ± 0.01** | 0.03 ± 0.003 | 0.51 ± 0.007 | N/A |
| CTGAN | 0.2 ± 0.02 | 0.14 ± 0.02 | 0.033 ± 0.006 | 0.22 ± 0.02 | N/A |
| Ours (S+V) | **0.17 ± 0.001** | 0.12 ± 0.02 | **0.027 ± 0.007** | **0.15 ± 0.02** | N/A |
| Ours (S) | 0.18 ± 0.01 | 0.13 ± 0.02 | 0.03 ± 0.007 | 0.19 ± 0.03 | N/A |

we observe a slight decrease in performance across datasets. Although the validator aims at improving the semantic integrity of the model, we see it also improves the general performance of the model as well.

Table 4 evaluates the performance of the generated synthetic data across different models when used for training in a classification task. Our method achieves remarkable results that are close to the performance of classifiers trained on real data, presenting the highest accuracy on the Adult, Covtype, Child and Intrusion datasets, exceeding the performance of real data for Intrusion (0.97 vs. 0.91) and Child (0.51 vs. 0.49) — which could be attributed to the generated dataset reducing the imbalance in the real dataset —, only reporting inferior performance for Census dataset compared to CTGAN (0.47 vs. 0.27), but the subpar KS and Chi Square test results of CATGAN for Census dataset suggest that it may have overfit to this particular classifier and problem, potentially generating data that lacks generalizability.

Table 3: Log Chi square test results for real data. We report the avarege KS test statistics for all categorical features of each dataset. Lower is better.

| Model | Adult ↓ | CovType ↓ | Intrusion↓ | Census ↓ | Child ↓ |
|---|---|---|---|---|---|
| MEDGAN | 9.4 ± 8.1 | 10.8 ± 10.8 | 8.6 ± 7.3 | 9.6 ± 8.1 | 8.4 ± 7.1 |
| TABLEGAN | 8.7 ± 7.4 | **5.4 ± 4.4** | **3.1 ± 2.2** | 9.1 ± 8. | 6.0 ± 5.6 |
| CTGAN | 8.1 ± 6.2 | 9.9 ± 7.1 | 9. ± 6.6 | 8.45 ± 6.9 | 6.4 ± 5.6 |
| Ours (S+V) | **7.8 ± 5.** | 14.5 ± 13. | 8.9 ± 5.2 | **8.38 ± 6.8** | **4.49 ± 3.1** |
| Ours (S) | 8.0 ± 4.9 | 10.1 ± 7.5 | 9 ± 5.5 | 10.5 ± 10.4 | 4.6 ± 3.12 |

Table 4: Downstream Efficacy results. We compare the downstream efficacy of the same model trained on real data and synthetic data. Higher is better.

| Model | Adult ↑ | Covtype↑ | Intrusion↑ | Census ↑ | Child ↑ |
|---|---|---|---|---|---|
| Real Data | 0.67 ± 0.0 | 0.82 ± 0.02 | 0.91 ± 0.07 | 0.51 ± 0.01 | 0.49 ± 0.0 |
| MEDGAN | 0.48 ± 0.3 | 0.11 ± 0.02 | 0.19 ± 0.08 | 0.03 ± 0.03 | 0.31 ± 0.08 |
| TABLEGAN | 0.24 ± 0.09 | 0.49 ± 0.02 | 0.79 ± 0.11 | 0.04 ± 0.01 | 0.46 ± 0.02 |
| CTGAN | 0.61 ± 0.02 | 0.46 ± 0.84 | 0.10 ± 0 | **0.47 ± 0.0** | 0.49 ± 0.01 |
| Ours (S+V) | **0.64 ± 0.0** | **0.59 ± 0.01** | **0.97 ± 0.0** | 0.27 ± 0 | **0.51 ± 0.0** |
| Ours (S) | 0.64 ± 0.0 | 0.57 ± 0.02 | 0.96 ± 0.0 | 0.27 ± 0.0 | 0.49 ± 0.0 |

## 5.5 Exploration Results

We start presenting the semantic integrity results by exploration factor (Table 5), with higher values indicating a greater diversity in the generated synthetic data. Our Method consistently achieves the highest exploration

factor across all datasets. The ablation model performs well but falls short of the model with validator component, suggesting that the validator contributes to the diversity of the generated data. The results indicate that Ours (S+V) not only generates data with higher fidelity but also with greater diversity, which is essential for robust synthetic data generation. However, the exploration factor alone is insufficient for a comprehensive assessment, as excessive diversity could result in the production of semantically inaccurate samples. Consequently, we extend our analysis to include the exploitation aspects of our method to ensure the synthetic data's semantic soundness.

Table 5: Exploration factor for different models on real datasets. We report the cumulative Hamming distance between a row in the sampled data and its closest counterpart in the original dataset. Higher exploration means more diversity in the generated samples which is desired.

| Model | Adult ↑ | Covtype↑ | Intrusion↑ | Census ↑ | Child ↑ |
|---|---|---|---|---|---|
| MEDGAN | $2240 \pm 43\%$ | $118 \pm 100\%$ | $2282 \pm 71\%$ | $132774 \pm 9\%$ | $1058 \pm 42\%$ |
| TABLEGAN | $6163 \pm 20\%$ | $81 \pm 43\%$ | $1090 \pm 27\%$ | $240295 \pm 11\%$ | $2381 \pm 33\%$ |
| CTGAN | $4296 \pm 13\%$ | $\mathbf{721 \pm 29\%}$ | $7849 \pm 27\%$ | $117534 \pm 6\%$ | $1392 \pm 13\%$ |
| Ours (S+V) | $\mathbf{14571 \pm 5\%}$ | $259 \pm 11\%$ | $\mathbf{17078 \pm 2\%}$ | $\mathbf{315871 \pm 4\%}$ | $\mathbf{2694 \pm 4\%}$ |
| Ours (S) | $10432 \pm 2\%$ | $141 \pm 18\%$ | $15922 \pm 4\%$ | $249433 \pm 1\%$ | $1374 \pm 2\%$ |

## 5.6 Exploitation

**Data**  To assess the semantic integrity of the synthetic data generated by our method, we focus on the *Child* dataset, which is our most challenging dataset - since it is only comprised of categorical features. We select two base features, the binary $X_1$ and the multivariate $X_2$ from the dataset. For each base feature, we generate two synthetic binary variables, i.e. $(V_1, V_2)$ associated with $X_1$ and $(V_3, V_4)$ associated with $X_2$. The synthetic variables $V_i$ are populated according to binomial distributions conditioned on the subgroups of their respective base features. Formally, let $V_i | X_{j,k} \sim \text{Binomial}(n_{j,k}, p_{i|j,k})$, where $n_{j,k}$ and $p_{i|j,k}$ are the parameters corresponding to $V_i$ given subgroup $X_{j,k}$.

We set the binomial distribution parameters $p_{i|j,k}$ to reflect intentional overrepresentation or underrepresentation in subgroups of base features. We also impose the condition that $p_{i|j,k} = 0$ for at least one $k$ in each $V_i$, ensuring that there exist subgroups within the base features that are deterministically assigned a synthetic value of 0, reflecting semantically invalid combinations.

As an example in a medical dataset, consider $X_1$ as patient sex, with $V_1$ indicating ovarian cancer presence and $V_2$ indicating prostate cancer presence. The probabilistic model mandates $V_1 | X_{1,\text{male}} \sim \text{Binomial}(n, 0)$ and $V_2 | X_{1,\text{female}} \sim \text{Binomial}(n, 0)$, thereby encoding the semantic constraints that men cannot develop ovarian cancer and women cannot develop prostate cancer directly into the synthetic data generation process.

**Results**  We consider the number of semantically incorrect samples (Table 6) as well as both the histogram of the positive samples for each variable (Figure 3) and the exploration factors from previous section in tandem to make sure that the model is not prohibiting semantically incorrect samples by reduced exploration and suppressing under-represented classes altogether. In Table 6 we report the percentage of smenatically incorrect samples for each model. For the sake of clarity, we use our model as baseline and report the difference for other models—i.e. negative values mean less violation of the semantic rules.

We first pay attention to MEDGAN and TABLEGAN since they seem to produce less semantically incorrect samples. Looking at statistical results (Table 3) reveals that both perform generally worse compared to our method. The results of efficacy in downstream task are also inferior to our method $(0.49, 0.31$ vs. $0.51)$. This gives an overall picture of the general performance of these two models not being good enough. Moving to the semantic aspects of MEDGAN and TABLEGAN's performance, we see that their exploration factor is lower than our model, suggesting that the lower count of semantically incorrect samples come at the cost of low diversity in the generated samples. This is highlighted in the subgroup level histogram, where for example MEDGAN suppresses subgroup 1 of $V2$. Thus, their superiority is at the cost of general performance (please note the overall inferiority of these two methods in estimating complex distributions for other datasets as well).

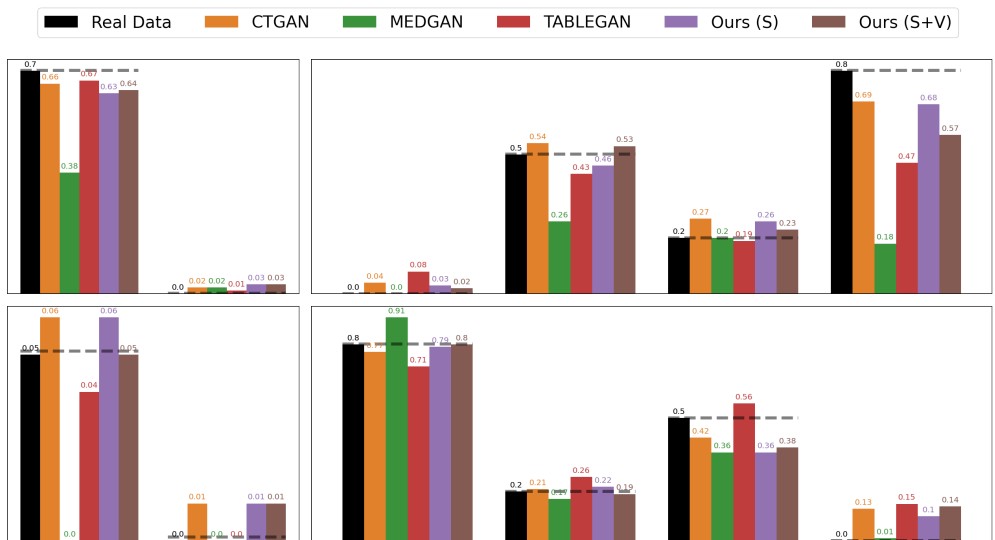

Figure 3: Histogram chart of subgroups for our synthetic variables $V1$ (top left), $V2$ (bottom left), $V3$ (top right) and $V4$ (bottom right) for the *Child* dataset. The real data values are shown in black. The plot shows which models suppress subgroups in their generated data. Suppressing features could have consequences for the synthetic dataset's general performance, fairness and semantic integrity.)

CTGAN, unlike MEDGAN and TABLEGAN, reports good overall performance in our experiments. Compared to CTGAN (which overall shows the best performance out of the compared models), our method shows better performance except for $V4$ where our model has a slight disadvantage of $-0.08\%$. That is while our method reports the highest exploration factor. The results of this section combined with general high performance and especially high exploration factor indicates superiority of our solution to the problem of unsupervised detection and enforcement of semantic bounds.

The ablation model consistently underperforms compared to the same GAN whose training was guided by the validator, leading us to conclude that the general superior performance of the synthetic data generated by our method both on general utility and specifically on semantic integrity is the result of the feedback from the latent space validator model.

We also note that the ablation model does not show a clear and dominant advantage compared to CTGAN, MEDGAN and TABLEGAN which leads us to the conclusion that these models that applying semantic boundaries on a relatively simple GAN will allow it perform generally better than the more sophisticated models tailor made to produce high quality synthetic data.

Table 6: Semantic integrity on Child dataset w.r.t. synthetic variables. We report percentage of samples with wrong labels. We use our method as the baseline and report the differences for other models.

| Model | $V1 \downarrow$ | $V2 \downarrow$ | $V3 \downarrow$ | $V4 \downarrow$ |
|---|---|---|---|---|
| MEDGAN | $-0.77 \pm 1.54$ | $-0.56 \pm 0.0$ | $-0.39 \pm 0.05$ | $-0.43 \pm 0.04$ |
| TABLEGAN | $-2.08 \pm 0.23$ | $-0.23 \pm 0.2$ | $2.09 \pm 0.9$ | $-0.14 \pm 0.12$ |
| CTGAN | $0.63 \pm 0.69$ | $0.01 \pm 0.21$ | $0.5 \pm 0.27$ | $-0.08 \pm 0.06$ |
| Ours (S+V) | $0.0 \pm 0.1$ | $0.0 \pm 0.1$ | $0.0 \pm 0.06$ | $0.0 \pm 0.06$ |
| Ours (S) | $0.2 \pm 0.06$ | $0.26 \pm 0.12$ | $0.27 \pm 0.05$ | $0.11 \pm 0.03$ |

# 6    Discussion and Future Work

In this work, we addressed the challenge of synthesizing tabular data with improved semantic integrity and reduced hallucination. We defined this task as an unsupervised constrained optimization problem. Our

approach introduces a novel, loosely-coupled architecture that combines a generic probabilistic synthesizer GM with a latent variable evaluator GM. This architecture aims to balance the exploration/exploitation trade-off by incentivizing the synthesizer to produce a diverse dataset while the validator learns semantic boundaries of the data in an unsupervised fashion and applies them as soft bounds during the synthesizer training to enhance semantic integrity. Going beyond the specific synthesizer and validator models presented in this paper, we argue that this framework could be extended to other latent space models, providing flexibility for researchers to improve the semantic integrity of their own data synthesis models. We defined new metrics specifically aimed to measure the exploration vs. exploitation of our proposed method as well as several state of the art tabular data synthesizers and showed overall improvements in general performance as well as semantic integrity.

Of note is our decision to condition the validator on all of the discrete variables to test our framework for the most challenging case. In real life, we expect only a subset of the features to be semantically important, which could improve performance by reducing the complexity of the semantic modelling process. Our framework allows for easy integration of several validators, each focused on a specific subset of categories and for the impact of each to be adjusted by the end user for fine grained control of exploration/exploitation aspect of the synthesizer.

For future work, one could focus on improving explainability of the validator component. Moving the validator from a blackbox model of the semantic boundaries in the latent space to an interpretable, explainable model would benefit the users in mission critical domains such as healthcare better understand the inner workings of the synthesizer, assess the quality of the generated samples and would enable them to incorporate their domain expertise into the validator to further improve the quality of the synthetic data.

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
