# OpenReview forum: "Synthesizing Tabular Data with Latent Semantic Regularization"
_TMLR — Rejected by TMLR_

### Review · Reviewer_FC9r · 2024-04-21

**Summary Of Contributions:**

This paper explores tabular data generation using generative models. They posit that generative models (GANs) do not produce samples that are 'semantically' consistent. This defect is remedied by adding an additional regularizer to the the data likelihood term. The regularizer is then designed as a VAE wherein the ELBO is minimized to minimize the distance between the learned posterior $q(z|x)$ and what they call the 'approximate' posterior $p(z|x)$. The VAE regularizer is termed the validator in the paper, and the claims are that it improves the GAN generation to get what the paper terms better exploration (sample diversity) and exploitation (sample quality).

Evaluations are carried out in a toy Gaussian mixture dataset with label assignment (presumably using the AC-GAN setup), and public tabular datasets.

**Audience:**

Yes

**Broader Impact Concerns:**

No concerns noted.

**Claims And Evidence:**

No

**Requested Changes:**

The paper needs a major revision for it to be comprehensible and make for meaningful reading. Some suggestions:
- Rewrite the methods section to include the auxillary classifier material. This is to motivate the ideas behind that work and inform the reader of the setting - I am not very sure about this point, especially as the validator component seems to have the conditioning variable $y$ which looks like a label.
- Bring up connections with other methods where such regularization is used - e.g. adversarial autoencoders [1].
- I find the clustering experiments quite confusing. Is the model trained to produce labels (auxillary classifier?).
- As mentioned in the section above, I find the emphasis on tabular data quite arbitrary. The authors should perhaps describe this in a little more detail.

Some minor points:
The first equation under section 3.1 for the validator ($\ln p(x|y) \cdots ...) has an extra comma in the term containing the KL divergence.
The long tailed posterior is referenced: 'anonymous. anonymous. In in submission, 2024.'  This should be rewritten.

**Strengths And Weaknesses:**

Strengths
+ The idea of enforcing semantic consistency is well motivated.
+ Tabular data in general is not as well explored as images or text.

Weaknesses
- I had a hard time understanding the method generally. The argument 'validator' component does not seem convincing.
- How is this different from a VAE-GAN setup such as the adversarial autoencoders work by Makhzani et al. [1] where the GAN generates synthetic data, and the VAE makes the latent space similar to the Gaussian?
- The design of the conditioning setup is not very cogent. For instance, equation in 3.1 for the validator containing the variable $y$. Is that the generated sample $x'$ or is it the conditioning label from AC-GAN?
- More unclear aspects: e.g. the statement following equation (2):  "cross-entropy between $q_\phi(z'|x) and the approximate posterior $p_\theta(z'|x')$ induced by $f$ when the generated samples $x'$ are passed through $q$". This appears simply incorrect to me - as far as I know it, $q(z|x)$ is the approximate posterior, while $p(z|x)$ is the real posterior which is intractable.
- The connection with tabular data seems weak. How does the method account for the particular features of tabular data.

In summary, I find the paper's claims not supported by a clear formulation, and there are many aspects where changes might be necessitated to make the paper more readable.


[1] https://arxiv.org/abs/1511.05644

---

> ### Author Response · Authors · 2024-05-12
> **Rebuttal**
>
> We thank reviewer FC9r for their useful feedback. Please find below our answer to your comments:
>
> The constrained optimization framework employs a validator component, crucial for ensuring semantic integrity by minimizing a loss function under explicit semantic constraints captured in the latent space. This setup enforces semantic norms by penalizing deviations, effectively managing semantic accuracy. The validator's design to address tail behavior of latent variables ensures the model explores data distribution extremes without semantic violations. The independence of the validator from the synthesizer prevents overfitting, maintaining unbiased semantic checks. Thus, this framework robustly upholds semantic integrity in synthesized data, enhancing model reliability across complex domains.
>
> Our method diverges from the VAE-GAN. While both methodologies integrate aspects of VAEs and GANs, the objectives and structural frameworks differ substantially in addressing data quality and semantic integrity.VAE-GAN trains a VAE with a dual objective: minimizing the traditional VAE loss while also employing a GAN-like adversarial training criterion. This criterion ensures that the aggregated posterior distribution of the latent representations aligns with a predetermined prior distribution, typically Gaussian. In contrast, our method uses a generic synthesizer (we used a GAN for our experiments, can be replaced with other types of generative models) and a latent space-based validator model (we use a VAE, can potentially be replaced with other types of latent space models) which operate in a loosely-coupled manner. The synthesizer is tasked with generating data. The VAE validator a)learns implicit semantic boundaries in its latent space and b) applies soft semantic bounds to ensure the semantic integrity of the generated data.
>
> The validator is actually a conditional VAE. We erroneously removed that detail while trying to trim the paper to be below the page limit. In paragraph 2 of discussion, we discuss the choice of $y$.
>
> We fixed the definition in section 3. It is actually the cross-entropy between the variational distribution and the prior, as is later explained in 3.1.
>
> Using tabular data allows us to test and evaluate whether the semantic constraints are satisfied. With other data types, such as images, this task is much harder to define let alone evaluate and measure.
>
> Your comment regarding the auxiliary classifier has been addressed in section 3.
>
> The connection between similar methods, including Makhzani et al. has been discussed in section 4.
>
> Since semantic integrity is challenging to evaluate, we had to come up with our own set of metrics and experiments. For this one, we took a standard clustering algorithm and trained a clustering model on it to generate for us the labels as the ground truth. Then we trained our models on this dataset(data+label) and generated samples from them. Then we fed the data from the generated samples - sans the label - to the original, already-trained, clustering method so it would predict their labels. We took these as true labels and compared them to the generated labels.
>
> In the context of semantic integrity in generative models, the structured nature of tabular data enables the easier evaluation of domain-specific rules. Each column in tabular data has a specific meaning and data type, sometimes with a truncated support, allowing for the enforcement of evaluation of logical constraints that reflect real-world conditions. For example, medical datasets can incorporate rules to prevent illogical entries such as a male patient listed as pregnant. These constraints are typically deterministic and can be rigorously defined and enforced. In contrast, defining semantic integrity for other data types, e.g. images, involves interpreting complex visual content where 'correctness' might relate to subjective or context-dependent interpretations such as the realism of synthetic images or the preservation of factual content in altered images. Please note that even for tabular data, we had to define new metrics and experiments to be able to evaluate our model. Our work is funded by a medical consortium which while sensitive to any type of semantic inconsistency, has prioritized tabular data above, for example, medical images. We perceive this as the beginning of an exciting line of research that, once extended in the domain of tabular data, could also be applied in other domains as well.

---

### Review · Reviewer_uzZH · 2024-04-27

**Summary Of Contributions:**

This work aims to improve the semantic integrity of generative models with tabular data.  Semantic integrity is defined to be explicit or implicit property of the data such as the domain knowledge or factual constraints.  A GAN for synthesis with the semantic constraints through the latents of the VAE is proposed to generate semantically consistent data. Experiments on different datasets show that the approach yields semantically better tables compared to prior GAN based approaches.

**Audience:**

Yes

**Broader Impact Concerns:**

There is no discussion on the broader impact of the work and would be required as part of the manuscript.

**Claims And Evidence:**

No

**Requested Changes:**

The related work is not appropriately discussed. As pointed out in the weakness, there is plethora of work at the intersection of VAEs and GANs which is not at all discussed in the manuscript. Similarly, well-established literature exists for using normalizing flows as priors in the latent space. This work does apply it to the tabular data but the technical novelty is limited and is claimed otherwise in the manuscript.
The design choice of using a GAN is not clear, the architectural or design choices are not ablated. All these concerns need to be critically addressed.

**Strengths And Weaknesses:**

+ The work aims to address the problem of semantically consistent tables in generative models.
+ The quantitative results show that the approach outperforms GAN based approaches for table generation.

- It is not clear why GANs have been used for synthesis? Why are VAEs not considered for generation directly?
- GANs and VAEs to improve the semantics in the latent space have been considered in various prior work such as [a,b,c]. The prior work in this domain should be included in the related work section
- Limited novelty: The technical novelty is limited. Even the normalizing flow priors are already heavily used in the latent space of VAEs. The important works in this direction are [d,e,f]. These works are not discussed. The overall framework and the idea of enhancing semantics with hybrid VAE-GANs exists and unfortunately is not all discussed in the manuscript
- Results shown in Table 6 are not clear. Why is lower better if the proposed model is considered as the mean or baseline? Should not the numbers closer to the baseline be the best?
- Equations are not numbered! Equation corresponding to L_v is not a loss.
- What is meant by the approximate posterior p_\theta(z'|x)? p_\theta corresponds to the synthesizer. How is this posterior computed?
- Missing ablations: There are no ablations to validate the working of the different components in the model.

[a] “Best-of-Many-Samples” distribution matching. Bhattacharyya et al. NeurIPS BDL, 2019.

[b]  Distribution Matching in Variational Inference.  Roska et al. 2019

[c] CVAE-GAN: fine-grained image generation through asymmetric training. Bao et al. CVPR 2017

[d] Latent Normalizing Flows for Discrete Sequences. Ziegler et al. ICML 2019

[e] Conditional Flow Variational Autoencoders for Structured Sequence Prediction. Bhattacharyya et al. NeurIPS BDL, 2019.

[f] Latent Normalizing Flows for Many-to-Many Cross Domain Mappings. Mahajan et al. ICLR 2020

---

> ### Author Response · Authors · 2024-05-12
> **Rebuttal**
>
> We thank reviewer uzZH for their insightful feedback. Please find below our answer to the points you raised:
>
> In this paper, we are proposing a modular approach to learn the semantic boundaries and to enforce them on the synthesizer. GANs have been utilized primarily for synthesis due to their proven ability to generate high-quality synthetic data, particularly in the domain of tabular data. GANs, such as CTGAN and TableGAN, have demonstrated superior performance in synthesizing tabular data, which supports their use in our framework. Furthermore, the architecture of our proposed system benefits from having an independent validator, which in our case is implemented as a VAE. However, the main emphasis of our approach is still the modular model-agnostic approach to solve the issue of the semantic integrity of the generated data. This separation of modules ensures that the validator can objectively evaluate the semantic integrity of the synthesized data without being influenced by the generation process of the synthesizer. An integrated model might risk the validator being too tightly coupled with the generator, potentially leading to overfitting where the synthesizer might exploit shared structures to minimize errors superficially without genuinely enhancing the semantic quality of the data.
>
> The paper has been modified to reflect your comments regarding previous works (section 4).
>
> In table 6, we show the amount of incorrectly labelled generated samples. We are showing the absolute difference of the performance of different models compared to our method. That is why lower than 0 numbers mean that method is generating less incorrectly labelled samples. The discussion about the nuances of interpreting these results are provided in the same section.
>
> Notations are updated for clarity (including regarding $L_v$). We will number the equations accordingly for the camera ready version so as to not confuse them in the revised version during the rebuttal period.
>
> The issue regarding $ p_\theta(z') $ has been fixed in the text.  We’re using the cross entropy between variational distribution and the prior as a measure of semantic correctness. Discussion about this choice is provided in Section 3.1, i.e. cross-entropy between posterior $ q_\phi(z'|x) $ and the prior $ p_\theta(z') $.
>
> The main focus of this work is the unsupervised modelling of semantic boundaries through the “validator” latent space model and then enforcing them on the synthesizer. Therefore, we provide ablation in form of reporting the results from our generic GAN synthesizer with and without the validator component. Since the focus of this paper is on improving semantic integrity of the output and reducing the likelihood of generating incorrect samples which could lead to the whole dataset becoming invalid (e.g. in medical domain), we believe that the provided ablation sufficiently reflects the impact of the validator. We’re open to any suggestions for additional ablations which would help shed light on the performance of our proposed method in the defined scope and will strive to include them in the camera ready version.

---

### Review · Reviewer_kydK · 2024-05-02

**Summary Of Contributions:**

This paper studies the problem of synthesizing tabular data has a high level of semantic integrity, which means that sample rows from the generated data should not have pairs of features that cannot be combined. The components of the model are a VAE validator and a GAN synthesizer. The GAN generator is responsible for generation from noise, while the validator encourages GAN samples to have better semantic integrity. Experiments on tabular data generation show the proposed method can achieve strong tabular synthesis results.

**Audience:**

No

**Claims And Evidence:**

No

**Requested Changes:**

I requested that the questions raised in the weakness section be answered and that details related to these questions are made clearer in revisions. In its current state, the technical details are difficult to follow and the specifics of the VAE/GAN interaction and the inclusion of semantic constraints in not clear.

**Strengths And Weaknesses:**

**Strengths**:
* Investigating methods to improve the semantic integrity of tabular data is an interesting problem with many useful applications.
* Experimental results show improved generation of tabular data compared to several prior methods.

**Weaknesses**:
* The technical details of the method are very difficult to follow. I was not able to clearly understand how the method works, and the notation was confusing in many places. The following bullet points list major sources of my confusion. I appreciate clarification from the authors in case of any significant misunderstandings on my part.
* *Equation (1)*: What is the relation between $x$ and $z$? It is confusing to write $p(x | z; \theta)$ without discussing their conditional or joint distribution. Why is the expectation be taken w.r.t. $p(x)p(z)$ rather than $p(x, z)$ (which would be a more typical Maximum Likelihood type objective)?
* *$f(x, z)$*: Why does the generator $f(x, z)$ require an observed data point $x$? Shouldn't the generator generate a new sample $x' = f(z)$ given only the latent signal? What is the relation between $x$ and $x'$?
* *Constraint $C(x)$*: A main theme of the paper is respecting constraints $C$ that measure semantic integrity. However, as far as I can tell, constraints $C(x)$ only appear in Eq. (1) and (2). How do the constraints affect the final objective at the end of Section 3.2? Are the constraints learned or hand-defined?
* *Validator*: How does the VAE affect the GAN? As far as I can understand, the VAE only affects the auxiliary loss term $L_v$ used to train the GAN. Is that accurate? It is difficult for me to see how the GAN parameters are even incorporated in $L_v$. Should the expectation be taken w.r.t. the GAN generator rather than the VAE encoder in the definition of $L_v$?
* Section organization could be improved, and extraneous discussion limited throughout the paper.

---

> ### Author Response · Authors · 2024-05-12
> **Rebuttal**
>
> We thank reviewer kydK for their insightful feedback. Please find below our answer to your comments:
>
> In Equation 1, the relationship between $x$ and $z$ is configured within a generative model framework where $z$ represents the latent variables that guide the generation of data samples $x$. The model is designed to optimize the generation process under the constraint $C(x')$, ensuring that all produced samples $x'$ meet specified semantic integrity constraints. $z$ influences the properties of $x$ through the conditional probability $p(x|z)$, balancing the need for diversity in data samples and adherence to semantic constraints.
>
> In Equation 1, the notation $ p(x \mid z; \theta) $ is used to denote the conditional probability of observing data $ x $ given latent variables $ z $, under a set of model parameters $ \theta $, as is typically employed in generative modeling contexts where the focus is on how data can be generated or reconstructed from a set of latent variables, emphasizing the model's ability to create or simulate data based on hidden representations.
>
> The reason for using $ p(x \mid z; \theta) $ without explicitly discussing the joint or conditional distribution involving $ z $ is to highlight the model's generative capabilities independently of how the latent variables are modeled or derived. This approach simplifies the exposition by focusing on the generative aspect of the model, which is crucial for tasks that involve data generation from learned latent representations.
>
> We separate $ x $ and $ z $ as independent entities under the assumption that $ x $ is sampled from the data distribution $ p_{\text{data}}(x) $ and $ z $ from some prior distribution $ p(z) $. This separation implies that $ x $ and $ z $ are treated as having independent sources of variability, which is a common approach in generative modeling to simplify the learning process by decoupling the generative factors. By treating $ x $ and $ z $ independently, the model simplifies the objective function. If $ p(x, z) $ were to be used, it would imply a joint distribution where the dependencies between $ x $ and $ z $ must be modeled explicitly. This would complicate both the model architecture and the learning algorithm, as it would require capturing the joint behavior directly. The objective function focuses on maximizing $ \log p(x|z;\theta) $, which aligns with learning how $ x $ can be generated given $ z $. This conditional generation is central to many generative models where $ z $ acts as a latent, controlling variable, and $ x $ is the output whose distribution conditioned on $ z $ we seek to model effectively. This setup supports the idea of $ z $ being a source of controlled variability for generating $ x $, rather than both being jointly modeled.
>
> The generator function is indeed $ f(z) $. This has been corrected in the paper. However the training objective requires a sample as an input to the validator component.
>
> We have updated the notations regarding $C(x)$ to reflect your feedback.
>
> The VAE significantly influences the GAN by modifying its loss function through the auxiliary loss term, which is constructed based on the output from the VAE encoder when processing samples generated by the GAN generator. The primary mechanism through which the VAE affects the GAN is by embedding the negative cross-entropy term into the GAN's generator objective function. This embedding is realized by first feeding the GAN-generated samples into the VAE encoder and then calculating the cross-entropy between the encoder's output distribution $q_\phi(z|x)$ and the prior distribution $p(z)$. The negative cross-entropy term effectively quantifies the divergence of the encoded distribution from the prior, thus providing a measure of how "unlikely" or "atypical" the generated samples are with respect to the assumed latent space distribution. By integrating this measure into the GAN's loss function, the generator is penalized for producing samples that deviate significantly from what is statistically expected, thereby reinforcing semantic integrity in the generated data.
>
> The expectation is indeed taken with respect to the distribution $q_\phi(z|x)$, which is the output of the VAE encoder when given a sample $x$ generated by the GAN. It is crucial to recognize that while the expectation in $\mathcal{L}_v$ is calculated over the distribution modeled by the VAE encoder, the parameters of the GAN generator influence $\mathcal{L}_v$ indirectly through the generation of the input samples $x$. Thus, the GAN generator's parameters are entwined with $\mathcal{L}_v$ because the quality and characteristics of the input $x$ directly affect the VAE encoder's output distribution.

---

### Decision · Action_Editor_uqTf · 2024-06-10

**Recommendation:** Reject

**Comment:**

This work aims to improve the semantic integrity of generative models with tabular data. After author rebuttal, this paper received 2 Leaning Reject and 1 Reject recommendations. All the reviewers shared one major concern that the technical details of the method are hard to follow and understand, and the authors also acknowledge numerous typos and confusions about major aspects of the original version of the paper. It will require significant efforts of revision to make the paper ready for publication. Besides this, reviewers also commented that (1) the related work is not appropriately discussed; as pointed out in the weakness, there is plethora of work at the intersection of VAEs and GANs which is not at all discussed in the manuscript; (2) the connection of the proposed method with tabular data seems weak. It's unclear wow the method accounts for the particular features of tabular data.

**Audience:**

Yes, but the paper itself needs to be significantly revised at the first hand.

**Claims And Evidence:**

No. The paper needs significant revision to make it ready for publication.